# Project20: Does continuity of care and community-based antenatal care improve maternal and neonatal birth outcomes for women with social risk factors? A prospective, observational study

**Hannah Rayment-Jones**[1]*, **Kathryn Dalrymple**[1], **James Harris**[2], **Angela Harden**[3], **Elidh Parslow**[4], **Thomas Georgi**[5], **Jane Sandall**[1]

**1** Department of Women and Children's Health, Faculty of Life Sciences & Medicine, King's College London, London, United Kingdom, **2** Clinical Research Facility, Chelsea and Westminster NHS Foundation Trust, London, United Kingdom, **3** School of Health Sciences, City University of London, London, United Kingdom, **4** St Mary's Hospital, Imperial College NHS Trust, London, United Kingdom, **5** School of Population Health & Environmental Sciences, Faculty of Life Sciences & Medicine, King's College London, London, United Kingdom

* Hannah.Rayment-Jones@kcl.ac.uk

## Abstract

### Background

Social factors associated with poor childbirth outcomes and experiences of maternity care include minority ethnicity, poverty, young motherhood, homelessness, difficulty speaking or understanding English, migrant or refugee status, domestic violence, mental illness and substance abuse. It is not known what specific aspects of maternity care work to improve the maternal and neonatal outcomes for these under-served, complex populations.

### Methods

This study aimed to compare maternal and neonatal clinical birth outcomes for women with social risk factors accessing different models of maternity care. Quantitative data on pregnancy and birth outcome measures for 1000 women accessing standard care, group practice and specialist models of care at two large, inner-city maternity services were prospectively collected and analysed using multinominal regression. The level of continuity of care and place of antenatal care were used as independent variables to explore these potentially influential aspects of care. Outcomes adjusted for women's social and medical risk factors and the service attended.

### Results

Women who received standard maternity care were significantly less likely to use water for pain relief in labour (RR 0.11, CI 0.02–0.62) and have skin to skin contact with their baby shortly after birth (RR 0.34, CI 0.14–0.80) compared to the specialist model of care.

**Data Availability Statement:** All relevant data are within the manuscript and its Supporting information files.

**Funding:** This report is independent research supported by the National Institute for Health Research (NIHR Doctoral Research Fellowship, HRJ, DRF-2017-10-033). AH is supported by the NIHR Applied Research Collaboration North Thames. JS (King's College London) is supported by the NIHR Applied Research Collaboration South London [NIHR ARC South London]. JS is also an NIHR Senior Investigator. JH is supported by a Post-doctoral Fellowship from Wellbeing of Women (Award Ref PRF006). KD is funded by the MRC (grant number: MR/V005839/1).

**Competing interests:** The authors have declared that no competing interests exist.

Antenatal care based in the hospital setting was associated with a significant increase in preterm birth (RR 2.38, CI 1.32–4.27) and low birth weight (RR 2.31, CI 1.24–4.32), and a decrease in induction of labour (RR 0.65, CI 0.45–0.95) compared to community-based antenatal care, this was despite women's medical risk factors. A subgroup analysis found that preterm birth was increased further for women with the highest level of social risk accessing hospital-based antenatal care (RR 3.11, CI1.49–6.50), demonstrating the protective nature of community-based antenatal care.

## Conclusions

This research highlights how community-based antenatal care, with a focus on continuity of carer reduced health inequalities and improved maternal and neonatal clinical outcomes for women with social risk factors. The findings support the current policy drive to increase continuity of midwife-led care, whilst adding that community-based care may further improve outcomes for women at increased risk of health inequalities. The relationship between community-based models of care and neonatal outcomes require further testing in future research. The identification of specific mechanisms such as help-seeking and reduced anxiety, to explain these findings are explored in a wider evaluation.

## Background and rationale

Health inequalities across the globe are influenced by social factors such as poverty, social deprivation, isolation, oppression, and discrimination. The large disparities seen in birth outcomes within high income countries often reflects their socioeconomic gradient, with mortality rates closely linked to disadvantages related to poverty, ethnicity, age and other social factors [1–6]. For example, the maternal mortality rate is disproportionality high for African American and Hispanic women in the US [7], Black, Asian and minority ethnic women in the UK [6, 8, 9], refugee and migrant women in other parts of Europe [10, 11], and Aboriginal and Torres Strait Islander women in Australia [12]. It is difficult to summarise the impact of specific social risk factors on inequalities in birth outcomes due to the nature of intersecting factors, and the accumulation of risk associated with poverty and ethnicity. Table 1 below presents an overview of social risk factors that are associated with poor birth outcomes and

**Table 1. Social risk factors associated with poor perinatal outcomes and experiences of maternity care [6, 7, 14, 18, 24–33].**

| Women who find services hard to access | Women needing multi-agency services |
|---|---|
| Black and Minority ethnicity | Mental health |
| Social isolation | Safeguarding concerns |
| Poverty/Deprivation/Homelessness | Substance and/or alcohol abuse |
| Refugees/Asylum seekers | Physical/emotional and/or learning disability |
| Non-native language speakers | Female genital mutilation |
| Victims of abuse | HIV positive status |
| Sex Workers | |
| Young Mothers | |
| Single Mothers | |
| Travelling community | |

exacerbate health inequalities in high income countries. Pregnancies of women with these social risk factors are over 50% more likely to end in stillbirth or neonatal death, and are associated with increased rates of miscarriage, termination, premature birth, low birth weight, caesarean section, and maternal death [6, 8, 13–23].

The UK ranks 22nd in maternal mortality for OECD countries [34], and 19th for infant mortality in Europe [35]. The London maternal mortality thematic review [15] found that over half of the 22 women who died in London in 2017 were from a Black or minority ethnic background, many had multiple complex social, medical and mental health factors. The review found that for the majority of maternal deaths there were missed opportunities to correctly diagnose and treat complications due to barriers across the maternity care pathway. The reviews recommendations are in line with the World Health Organisations stance on improving maternity care: *'To improve maternal health, barriers that limit access to quality maternal health services must be identified and addressed at both health system and societal levels'* [36].

There is a strong evidence base that good quality midwifery care, particularly when it involves continuity of care, leads to improved outcomes for women and children and the unnecessary use of interventions in high income countries [37, 38]. Midwife-led continuity of care is defined as when "the midwife as the lead professional in the planning, organisation and delivery of care given to a woman from initial booking to the postnatal period" [39]. Although the Cochrane review of models of midwifery care [40] found that women who received continuity of care had improved birth outcomes, fewer preterm births, fetal loss and neonatal death than those receiving standard maternity care, it did not report on whether outcomes differed for women with social risk factors. The authors recommended that future research should explore this population and address the underlying mechanisms of the improved outcomes. For example, whether the observed benefits can be attributed to the quality of the relationship between the midwife and woman, or other factors such as place of care. Other specialist models of maternity care, for example group antenatal care such as 'centring pregnancy' and 'pregnancy circles', and family nurse partnerships are currently being trialled to explore their impact on outcomes for women with social risk factors [41–45]. It is hypothesised that culturally safe and community-based models of care which adopt a life course approach might help to reduce maternal and neonatal health inequalities, enhance care and improve women's experiences of maternity care [46–48]. This impact of these place-based aspects of maternity care is poorly understood and under-researched, particularly in the UK context and for women with social risk factors who are more likely to be socially isolated and struggle to integrate with their local community.

There are a number of services across the UK providing 'specialist care' to women with social risk factors that often incorporate continuity of care in community settings and aim to reduce health inequalities, but they are under evaluated and often vulnerable to organisational restructuring [49]. Recent UK policy [50, 51] has focused on targeting access to continuity models of care for women living in deprived areas and those from Black, Asian and minority ethnic groups [52, 53]. However, there is a significant knowledge gap around the mechanisms of continuity of care and specialist models. It is not known how and why some models of maternity care appear more effective than others, or if the positive outcomes reported in the literature are experienced by Black, Asian and minority ethnic women, and those with social risk factors. An expert panel in maternal and newborn health research, including service user representatives set global research priorities for the reduction of maternal and perinatal mortality, and preterm birth and stillbirths [54]. The top research priorities included 'the evaluation of the effectiveness of midwifery care on access to family planning services, and rates of neonatal death, preterm birth and low birthweight'. Evaluating different models of care and

identifying the impact of factors such as continuity of care and where antenatal care is received will help inform the organisation of future services for this 'at risk' population.

## Aim and objectives

**Aim.**   To describe and compare maternal and neonatal clinical birth outcomes according to the model of maternity care women receive, and where their antenatal care is located.

**Objectives.**   By comparing outcome data the analysis will explore whether standard maternity care, group practice or specialist models affect:

- Maternal and neonatal birth outcomes, the use of pharmacological analgesia and obstetric interventions.

- Women's antenatal admissions to hospital and the length of their postnatal stay.

   The analysis will also seek to identify:

- Sociodemographic characteristics associated with maternal and neonatal outcomes.

- Whether the location of antenatal care has an additional effect on maternal and neonatal outcomes?

## Methods

### Study design

The analysis reported in this paper is from a wider multi-site prospective observational study evaluating two UK based specialist models that provide maternity care to women with social risk factors. Demographic data for the first 500 women accessing maternity care in January 2019 at two large, inner-city maternity services were prospectively collected and anonymised. Pregnancy and birth outcome data were collected and analysed in August 2019 for women who had gone on to give birth at one of the two maternity services being evaluated. Exclusion criteria included those who experienced miscarriage (loss of pregnancy during the first 23 weeks), or who had not continued their antenatal care at the service. Three different models of maternity care with varying levels of continuity, and place of antenatal care were used as independent variables to explore their impact on pregnancy and birth outcomes. The research was approved by the London Brent Research Ethics Committee (HRA) REC Reference 18-LO-0701. This study is reported as per the STROBE checklist for observational studies [55].

### Setting

Two inner city National Health Service (NHS) maternity service providers in the UK that provide care to a multi-cultural, socioeconomically diverse population were purposively selected. As current policy and literature on improving maternal and neonatal health inequalities recommends relational continuity of care [40, 30, 56, 57], we selected providers that offered well-established specialist models of care aiming to provide continuity throughout the antenatal, intrapartum and postnatal period, as well as standard maternity care; and group practice to allow for comparisons between the three. Table 1 in S1 Appendix provides detailed definitions of the two service provider settings and the different models of care which women might experience at each:

### Data sources and variables

Outcome variables were collected based on the availability and comparability of compulsory data recorded within each service providers computerised data collection programmes 'Cerner' and 'Badgernet'. To address potential sources of bias the data were collected and anonymised by clinicians outside of the research team.

Deprivation deciles, calculated using the 2019 English Indices of Deprivation [58] were grouped into four groups of sufficient numbers to enable comparisons between groups of similar numbers. These groups will be used throughout the findings chapters and are as follows:

1. Most deprived- 1st and 2nd deciles

2. 3rd and 4th deciles

3. 5th and 6th deciles

4. Least deprived- 7th, 8th, 9th and 10th deciles

Table 2 shows variables collected at each time point that will be presented in this paper. See Table 2 in S1 Appendix for definitions of variables.

### Sample size

A power calculation was based on a previous analysis of antenatal care utilisation in the UK [59] and research carried out on metrics for monitoring local inequalities in access to maternity care at the same service evaluated in this research [60]. This was due to the primary outcome of the full evaluation being access and engagement with maternity care. We calculated that with 250 women in each group (those receiving standard maternity care and those receiving group practice or a specialist model of care), we would have 90% power to detect a 15% difference in timely access (before 12 weeks' gestation) to antenatal care between the different models of care with 500 anonymised birth records accessed at each trust. As the study was not primarily powered to detect differences in the maternal and neonatal birth outcomes analysed for this paper, a retrospective power analysis using previous literature on the relationship between premature birth and specialist maternity care was calculated [38, 61] demonstrating the sample size was underpowered to detect previously reported differences. Despite this, statistical significance was identified for numerous outcomes, highlighting the usefulness of this analysis.

See Fig 1 for the data collection flowchart. Full pregnancy and birth data were collected and analysed from 799 women accessing care across the two service providers. Two hundred and one sets of birth outcome data were missing and were therefore excluded from the final analysis. Reasons for this missing data are reported in the findings.

### Statistical methods

The quantitative data were analysed using Stata 16.0. Firstly, women's social risk factors, ethnicity, socioeconomic status and medical characteristics were described using descriptive statistics and stratified by the service provider attended to enable comparisons of differences in the samples between each service. It was decided to merge the two service providers outcome data for ease of interpreting the findings, with adjustment allowing for differences between the service providers. Variables were tested for bivariate association using chi-square tests and $t$-tests, for dichotomous and continuous variables, respectively. Chi-square analyses were also performed to test for associations between socio-economic position by deprivation (IMD) decile [58], as well as social and medical risk factors.

**Table 2. Outcome variables collected at two time points.**

| Outcome variable | 1st data collection- January 2019 | 2nd data collection- August 2019 |
|---|---|---|
| **Characteristics** | | |
| Deprivation score | x | |
| Maternal age | x | |
| Ethnicity | x | |
| Parity | x | |
| Social risk factors (listed) | | x |
| No. of social risk factors | | x |
| Medical risk status at booking | x | |
| Medical risk status at onset of labour | | x |
| **Service use** | | |
| Reason if sample drop out | | x |
| Model of care received | | x |
| Place of birth | | x |
| Neonatal unit admission | | x |
| Length of postnatal stay | | x |
| **Birth outcomes** | | |
| Mode of birth | | x |
| Induction of labour | | x |
| Monitoring (CTG in labour) | | x |
| Perineal trauma req suturing | | x |
| Estimated blood loss | | x |
| Analgesia | | x |
| Obstetric emergency | | x |
| Maternal death | | x |
| **Neonatal Outcomes** | | |
| Sex | | x |
| Gestation at birth | | x |
| Weight | | x |
| Stillbirth/neonatal death | | x |
| Apgar scores | | x |
| Skin-to-skin | | x |
| Feeding method | | x |
| **Discharge information** | | |
| Date discharged home | | x |
| Social care involvement | | x |
| Baby discharged home with parents/ LAC | | x |

Three regression models were developed to identify the differences in the effect size for each: Model 1 adjusted for ethnicity, age, parity, deprivation score, social risk factors and medical risk status, Model 2 included model 1, plus adjustment for the service provider that women attended to consider differences in organisation guidelines, processes and culture and Model 3 included model 2, however, the place of antenatal care (hospital versus community-based antenatal care) was treated as the independent variable. This structured model allowed us to explore the association between maternal and neonatal outcomes depending on the model of care received, whilst accounting for interactions between independent variables to predict the dependent variable. A subgroup analysis of statistically significant findings was also conducted for those women who are at highest risk of poor birth outcomes. Where pregnancy

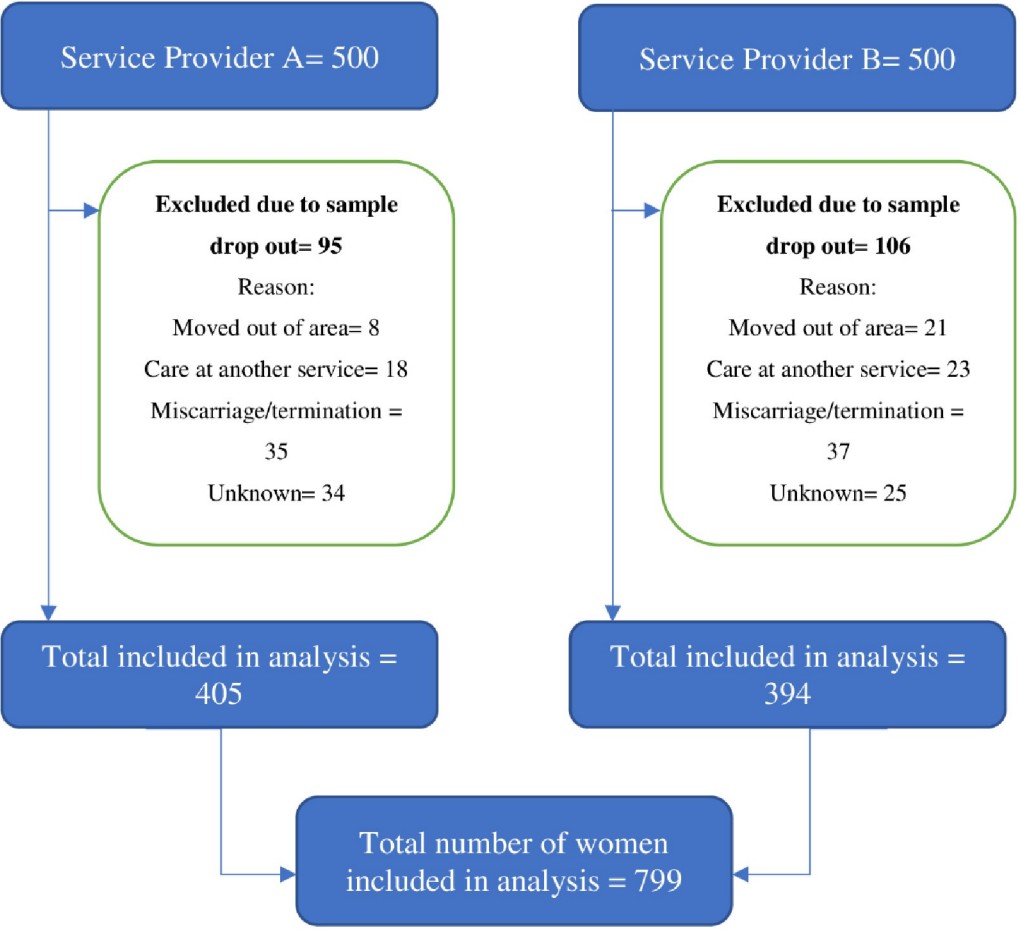

**Fig 1. Data collection flowchart.**

and birth outcomes are presented risk ratios and confidence intervals are used to demonstrate statistical significance as well as the direction and strength of the effect [62].

## Results

The section below describes the characteristics of the women in the quantitative sample. The women who were excluded due to drop out are presented first to explore differences between the two service providers and deprivation groups. P values are presented to show statistical significance between the characteristics of women accessing maternity care at both providers.

### Characteristics of women with missing outcome data

Of the first 1000 women who had an appointment to book for maternity care in 2019, 201 did not go on to give birth at the service and are not included in the quantitative data analysis that follows. The total numbers of women with missing outcome data at each hospital did not differ significantly, allowing the data from both services to be pooled without having to adjust for missing data. No significant difference was found in the reason for missing data when analysed according to women's deprivation scores.—See Tables 1 and 2 in S2 Appendix. The small sample size here should be kept in mind; an apparent trend may have become significant with a

larger sample size, reflecting the literature demonstrating a social gradient for both spontane-ous miscarriage and termination of pregnancy rates [13, 14, 21–23].

### Demographics of women included in the quantitative data analysis

The section below describes the demographics of the 799 women who went on to receive ante-natal care and give birth at one of the two service providers. Table 3 shows that more women at service A were recorded as 'white British', and more women at service B were recorded as 'white other'. Ethnicity was also more likely to be recorded as 'unknown' at service A. Women at service B were more likely to have at least one social risk factor recorded, have common mental health issues, drug and/or alcohol abuse, financial and/or housing issues, be non-English speaking, unsupported, and have disclosed female genital mutilation (FGM). Women at service B were also significantly more likely to have high medical risk status at the onset of labour. Similar numbers of women experienced standard care and private (Non-NHS) care at both service providers. However, more women at service B received the group practice model, and more women at service A received specialist models of care. More women at service B experienced standard care in the hospital setting whereas more women at service A experience standard care based in the community setting. Women receiving private care were not included in the analysis as private care is not a realistic option for women with low socioeco-nomic status, and numbers were too small to gain generalisable learning. The differences reported here informed the modelling structure that adjusted for women's demographics and risk factors.

Significant differences were found in the care received by women depending on their depri-vation score- See Table 3 in S2 Appendix. Women in the most deprived deciles were signifi-cantly more likely to receive a specialist model of care, and women in the least deprived deciles were less likely to receive community based antenatal care than hospital based antenatal care. A statistically significant relationship was also found between deprivation score and the num-ber of social risk factors recorded, reflecting the literature showing the lower a woman's socio-economic status, the more likely she is to be experiencing one of more social risk factors [63–67].

### Outcome data

#### Maternal birth outcomes

**Analysis 1- model of care.** The data presented in Table 4 tests the hypothesis that mater-nal birth outcomes will vary according to model of care. No significant relationship was found between the model of care received and women's birth outcomes, including mode of birth, blood loss, perineal trauma requiring suturing, and obstetric emergencies, after adjusting for women's characteristics and service differences. When adjusting for women's characteristics- See Table 3 in S2 Appendix, null parity was found to be a significant predictor of increased emergency caesarean section (RR4.96 CI 3.09–7.94) and instrumental delivery (RR 8.06 CI 4.71–13.79). Women at high medical risk at booking (RR 5.52 CI 2.20–13.83) and at the onset of labour (RR 2.66 CI 1.47–4.81) were more likely to have an elective or emergency caesarean, and instrumental delivery compared to women with low medical risk. Primiparous women were more likely to have a postpartum haemorrhage (PPH) (RR 3.23 CI 2.32–4.50), perineal trauma requiring suturing (RR 2.30 CI 1.67–3.17) and experience an obstetric emergency (RR1.94 CI 1.34–2.79). Women with high medical risk status at the onset of labour were also more likely to have a postpartum haemorrhage (PPH) (RR 1.84 CI 1.26–2.69), other obstetric emergency (RR1.77 CI 1.15–2.75), and less likely to have perineal trauma requiring suturing (RR 0.47 CI 0.32–0.69). A significant relationship was found between women with any social

**Table 3. Women's demographics at each service provider.**

| Demographic variable | Service A n(%) Total data = 405 | Service B n(%) Total data = 394 | TOTAL n(%) Total data = 799 | X² p value |
|---|---|---|---|---|
| **Ethnicity** | | | | Pr 0.000 |
| Asian | 37(9) | 53(13) | 90(11) | |
| Black African | 31(8) | 46(12) | 77(10) | |
| Black Caribbean | 23(6) | 16(4) | 39(5) | |
| Black other | 8(2) | 14(4) | 22(3) | |
| Mixed | 12(3) | 7(2) | 19(2) | |
| White British | 98(24) | 58(15) | 156(20) | |
| White other | 80(20) | 139(36) | 219(27) | |
| Unknown | 116(29) | 61(15) | 177(22) | |
| **Age** | | | | Pr 0.356 |
| ≤20 | 6(1) | 4(1) | 10(1) | |
| 21–24 years | 19(5) | 32(8) | 51(6) | |
| 25–29 years | 63(16) | 56(14) | 119(15) | |
| 30–34 years | 134(33) | 125(32) | 259(32) | |
| ≥35 years | 183(45) | 177(45) | 360(45) | |
| **IMD Quintile (2019)** | | | | Pr 0.035 |
| Most deprived (1st +2nd deciles) | 92(23) | 114(29) | 206(26) | |
| 3rd and 4th deciles | 160(40) | 126(32) | 286(36) | |
| 5th and 6th deciles | 72(18) | 86(22) | 158(20) | |
| Least deprived (7th, 8th, 9th +10th deciles) | 81(20) | 68(17) | 149(19) | |
| **Social Risk Factor** | | | | |
| Domestic abuse | 23(6) | 17(4) | 40(5) | Pr 0.377 |
| Common mental health | 4(1) | 34(9) | 38(5) | **Pr 0.000** |
| Severe mental health | 2(<1) | 8(2) | 10(1) | Pr 0.051 |
| Non-English speaking | 16(4) | 48(13) | 64(8) | **Pr 0.000** |
| Social care involvement | 27(7) | 29(7) | 56(7) | Pr 0.701 |
| Drug/alcohol abuse | 1(<1) | 10(3) | 11(1) | **Pr 0.005** |
| Unsupported/single | 1(<1) | 11(3) | 12(2) | **Pr 0.003** |
| Financial/housing | 15(4) | 31(8) | 46(6) | **Pr 0.012** |
| Learning disability | 6(2) | 5(1) | 11(1) | Pr 0.797 |
| Sexual abuse/trafficked | 4(2) | 5(1) | 9(1) | Pr 0.677 |
| AS/Refugee | 8(2) | 7(2) | 15(2) | Pr 0.836 |
| FGM | 0 | 11(3) | 11(1) | **Pr 0.001** |
| No recourse to public funds | 6(1) | 0 | 6(1) | **Pr 0.015** |
| **No of social risk factors** | | | | **Pr 0.003** |
| None | 337(83) | 279(70) | 616(77) | |
| 1 | 43(11) | 61(15) | 104(13) | |
| 2 | 13(3) | 26(7) | 39(5) | |
| 3 | 6(1) | 15(4) | 21(3) | |
| 4 | 5(1) | 9(2) | 14(2) | |
| ≥5 | 1(<1) | 4(1) | 5(1) | |
| **Name of model of care** | | | | **Pr 0.000** |
| Standard Care | 256(63) | 213(54) | 469(59) | |
| Group Practice | 77(19) | 144(37) | 221(28) | |
| Specialist | 59(15) | 21(5) | 80(10) | |
| **Private Care** | 13(3) | 16 (4) | 29(4) | |

(*Continued*)

**Table 3.** (Continued)

| Demographic variable | Service A n(%) | Service B n(%) | TOTAL n(%) | X² p value |
|---|---|---|---|---|
| | Total data = 405 | Total data = 394 | Total data = 799 | |
| **Place of model of antenatal care** | | | | Pr 0.000 |
| Standard model in hospital | 100(25) | 212(54) | 312(39) | |
| Standard model in community | 156(40) | 1(0) | 157(20) | |
| Group practice in community | 40(10) | 94(24) | 134(17) | |
| Group practice in hospital | 37(9) | 50(13) | 87(11) | |
| Specialist model in community | 59(15) | 2(1) | 61(8) | |
| Specialist model in hospital | 0 | 19(5) | 19(2) | |
| Private Care | 13(3) | 16(4) | 29(4) | |
| **By place of antenatal care only*** | | | | Pr 0.000 |
| Hospital based | 137(35) | 281(74) | 418(54) | |
| Community based | 255(65) | 97(26) | 352(46) | |

*Excludes private care.

**Table 4.** Maternal birth outcomes and model of care.

| Birth outcome | Model of Care | No. of women (%) | Unadjusted RR (95% CI) | Model 1 Adjusted RR (95% CI) * | Model 2 Adjusted RR(95% CI) ** | Model 3 Adjusted RR(95% CI) *** |
|---|---|---|---|---|---|---|
| Spontaneous vaginal birth | Standard | 209(54) | Ref | Ref | Ref | Ref |
| | Group | 132(34) | Ref | Ref | Ref | Ref |
| | Specialist | 44(12) | Ref | Ref | Ref | Ref |
| Instrumental delivery | Standard | 82(67) | 1.91(0.89–4.10) | 1.75(0.75–4.09) | 1.59(0.66–3.81) | 1.60(0.67–3.83) |
| | Group | 32(26) | 1.18(0.40–1.45) | 1.23(0.49–3.04) | 1.20(0.48–2.98) | 1.18(0.46–2.99) |
| | Specialist | 9(7) | Ref | Ref | Ref | Ref |
| Emergency caesarean section | Standard | 91(59) | 1.00(0.55–1.81) | 0.97 (0.48–1.95) | 0.91(0.44–1.86) | 0.91(0.44–1.88) |
| | Group | 44(29) | 0.77(0.40–1.45) | 0.33 (0.33–1.46) | 0.68(0.32–1.44) | 0.65(0.30–1.40) |
| | Specialist | 19(12) | Ref | Ref | Ref | Ref |
| Elective caesarean section | Standard | 87(81) | **2.28(1.03–5.06)** | 2.28(0.94–5.51) | 1.91(0.77–4.72) | 2.00(0.80–4.99) |
| | Group | 13(12) | 0.54(0.21–1.39) | 0.47(0.16–1.33) | 0.43(0.15–1.24) | 0.36(0.12–1.05) |
| | Specialist | 8(7) | Ref | Ref | Ref | Ref |
| Blood loss>500mls (PPH) | Standard | 249(64) | 1.25(0.77–2.01) | 1.07(0.63–1.82) | 1.02(0.59–1.76) | 1.02(0.59–1.76) |
| | Group | 102(26) | 0.94 (0.56–1.58) | 0.91(0.51–1.61) | 0.90(0.51–1.59) | 0.76(0.42–1.37) |
| | Specialist | 38(10) | Ref | Ref | Ref | Ref |
| Blood loss> 1000mls (MOH) | Standard | 40(65) | 0.73(0.34–1.58) | 0.88(0.38–2.03) | 1.00(0.42–2.36) | 0.99(0.41–2.34) |
| | Group | 13(21) | 0.49(0.20–1.20) | 0.57(0.22–1.48) | 0.59(0.22–1.54) | 0.69(0.26–1.83) |
| | Specialist | 9(15) | Ref | Ref | Ref | Ref |
| Perineal trauma req suturing | Standard | 199(60) | 1.22 (0.75–2.00) | 1.17(0.68–2.02) | 1.11(0.63–1.95) | 1.11(0.63–1.94) |
| | Group | 101(31) | 1.40 (0.83–2.36) | 1.47(0.82–2.64) | 1.45(0.81–2.61) | 1.38(0.76–2.51) |
| | Specialist | 30(9) | Ref | Ref | Ref | Ref |
| Obstetric emergency | Standard | 119(63) | 1.17(0.66–2.07) | 1.14(0.62–2.10) | 1.20(0.64–2.25) | 1.21(0.65–2.25) |
| | Group | 53(28) | 1.09(0.59–2.02) | 1.19(0.62–2.29) | 1.22(0.63–2.34) | 1.29(0.66–2.51) |
| | Specialist | 18(9) | Ref | Ref | Ref | Ref |

* Model 1: Adjusted for demographics ethnicity, age, parity, IMD score, any social risk factor and medical risk factors at booking and onset of labour.

** Model 2: Model 1 + Adjustment for place of antenatal care (community or hospital).

*** Model 3: Model 2 + Adjustment for service provider attended (A or B).

**Table 5. Maternal birth outcomes in relation to the place of antenatal care.**

| Birth outcome | Place of antenatal care | Number of women n (%) | Unadjusted RR (95% CI) | Model 1 Adjusted RR (95% CI) * | Model 2 Adjusted RR (95% CI) ** | Model 3 Adjusted RR (95% CI) *** |
|---|---|---|---|---|---|---|
| Spontaneous vaginal birth | Hospital | 187(49) | Ref | Ref | Ref | Ref |
| | Community | 198(51) | Ref | Ref | Ref | Ref |
| Instrumental delivery | Hospital | 69(56) | 1.34(0.89–2.02) | 1.43(0.90–2.26) | 1.27(0.78–2.07) | 1.24(0.74–2.10) |
| | Community | 54(44) | Ref | Ref | Ref | Ref |
| Emergency caesarean | Hospital | 88(57) | 1.38(0.95–2.02) | 1.27(0.82–1.97) | 1.19(0.75–1.89) | 1.12(0.68–1.83) |
| | Community | 66(43) | Ref | Ref | Ref | Ref |
| Elective caesarean | Hospital | 74(69) | **2.26(1.43–3.55)** | **2.10(1.26–3.48)** | 1.61(0.92–2.80) | 1.06(0.56–2.01) |
| | Community | 34(31) | Ref | Ref | Ref | Ref |
| PPH (Blood loss>500mls) | Hospital | 227(58) | **1.37(1.03–1.82)** | 1.16(0.84–1.60) | 1.11(0.79–1.55) | 0.92(0.64–1.33) |
| | Community | 162(42) | Ref | Ref | Ref | Ref |
| MOH (Blood loss> 1L) | Hospital | 30(52) | 0.77(0.46–1.30) | 0.76 (0.43–1.34) | 0.71(0.39–1.30) | 0.88(0.46–1.69) |
| | Community | 30(48) | Ref | Ref | Ref | Ref |
| Perineal trauma req suturing | Hospital | 155(47) | 0.92(0.69–1.23) | 1.08(0.78–1.49) | 1.15(0.82–1.61) | 1.08(0.75–1.56) |
| | Community | 175(53) | Ref | Ref | Ref | Ref |
| Obstetric emergency | Hospital | 89(47) | 0.91(0.66–1.27) | 0.89(0.62–1.27) | 0.85(0.58–1.24) | 0.91(0.61–1.37) |
| | Community | 101(53) | Ref | Ref | Ref | Ref |

* Model 1: Adjustment for demographics ethnicity, age, parity, IMD score, social risk and medical risk factors at booking and onset of labour.

** Model 2: Model 1 + adjustment for model of care.

*** Model 3: Model 2 + adjustment for service provider attended.

risk factor and massive obstetric haemorrhage (MOH) (RR 1.99 CI 1.03–3.83). Maternal death was not included in the analysis as numbers were too small to detect a relationship (n = 1).

**Analysis 2- place of antenatal care.** A second analysis was run on the impact of place of antenatal care on birth outcomes. Table 5 shows that, after adjusting for potential confounders, there was no significant relationship between the place of antenatal care and maternal birth outcomes.

### Analgesia in labour and obstetric interventions

**Analysis 1- model of care.** Table 6 shows that the only statistically significant relationship with model of care across all unadjusted and adjusted models was the use of water in labour. Women receiving the specialist model of care were most likely to use water to relive pain during labour, with those receiving standard care being least likely (RR 0.11 CI 0.02–0.62). When adjusting for women's characteristics those with high medical risk status at onset of labour (RR4.57 CI 2.97–7.503) and those over 34 years old (RR 5.85 CI 1.39–24.55) were significantly more likely to have an epidural. Primiparous women were most likely to have an epidural (RR 0.55 CI 0.37–0.82) and opioid analgesia (RR 4.81 CI 1.19–19.35). Differences seen in the number of women having a CTG in labour was largely driven by primiparous women (RR1.68 CI 1.06–2.64) and those with high medical risk status at the onset of labour (RR3.06 CI 1.94–4.83).

**Analysis 2- place of antenatal care.** Table 7 shows no significant relationship between the place of antenatal care and use of analgesia. However a significant relationship was found for women receiving antenatal care in the hospital being less likely to experience an induction of

**Table 6. Use of analgesia in labour and obstetric interventions in relation to the model of care received.**

| Analgesia in labour/ Intervention | Model of Care | Number of women (%) | Unadjusted RR (95% CI) | Model 1 Adjusted RR (95% CI) * | Model 2 Adjusted RR (95% CI) ** | Model 3 Adjusted RR (95% CI) *** |
|---|---|---|---|---|---|---|
| Epidural/CSE/GA | Standard | 306(64) | 1.30(0.80–2.11) | 0.99(0.56–1.73) | 1.01(0.57–1.80) | 1.01(0.57–1.80) |
| | Group | 123(26) | 0.89(0.53–1.49) | 0.72(0.39–1.32) | 0.73(0.40–1.33) | 0.71(0.38–1.31) |
| | Specialist | 47(10) | Ref | Ref | Ref | Ref |
| Opioid analgesia | Standard | 9(60) | 0.76(0.16–3.59) | 0.55(0.10–2.89) | 0.54(0.09–3.17) | 0.51(0.87–3.05) |
| | Group | 4(27) | 0.71(0.12–4.00) | 0.56(0.08–3.53) | 0.56(0.08–3.53) | 0.29(0.03–2.51) |
| | Specialist | 2(13) | Ref | Ref | Ref | Ref |
| No analgesia or Entonox | Standard | 90(53) | 0.62(0.36–1.07) | 0.70(0.38–1.29) | 0.74(0.30–1.38) | 0.73(0.39–1.37) |
| | Group | 57(34) | 0.96(0.51–1.62) | 1.00(0.52–1.90) | 1.01(0.53–1.93) | 1.15(0.59–2.22) |
| | Specialist | 22(13) | Ref | Ref | Ref | Ref |
| Water in labour | Standard | 3(23) | **0.09(0.02–0.41)** | **0.10(0.02–0.53)** | **0.14(0.02–0.72)** | **0.11(0.02–0.62)** |
| | Group | 5(38) | 0.34(0.09–1.23) | 0.48(0.10–2.22) | 0.50(0.10–2.31) | 0.65(0.14–3.06) |
| | Specialist | 5(38) | Ref | Ref | Ref | Ref |
| CTG in labour | Standard | 168(60) | **2.28 (1.27–4.07)** | 1.17(0.86–3.39) | 0.96(0.45–2.02) | 0.92(0.38–2.19) |
| | Group | 97(34) | **3.15(1.71–5.80)** | **2.69(1.31–5.48)** | **2.84 (1.34–6.01)** | 0.80(0.32–2.01) |
| | Specialist | 16(6) | Ref | Ref | Ref | Ref |
| Induction of labour | Standard | 203(60) | 0.89(0.55–1.43) | 0.89(0.52–1.52) | 1.10(0.63–1.91) | 1.10(0.63–1.91) |
| | Group | 97(29) | 0.90(0.54–1.51) | 0.85(0.48–1.51) | 0.90(0.50–1.61) | 1.01(0.56–1.83) |
| | Specialist | 37(11) | Ref | Ref | Ref | Ref |

* Model 1: Adjusted for demographics ethnicity, age, parity, IMD score, any social and medical risk factors at booking and onset of labour.

** Model 2: Model 1 + Adjustment for place of antenatal care (community or hospital).

*** Model 3: Model 2 + Adjustment for service provider attended (A or B).

labour (RR0.65 CI 0.45–0.95). The differences in the use of water for pain relief in labour were driven by the significant relationship with the model of care received.

## Place of birth

**Analysis 1- model of care.** Table 8 shows that overall, there was no significant difference between the model of care and place of birth. However, women attending service provider B were significantly more likely to give birth on the labour ward (RR 4.15 CI 2.46–7.00). See Table 19 in S2 Appendix for fully adjusted outcome tables.

**Analysis 2- place of antenatal care.** Table 9 shows no significant relationship between place of antenatal care and place of birth once the model adjusted for the service attended.

## Neonatal outcomes

**Analysis 1- model of care.** Table 10 shows no significant relationship between the model of care received and neonatal outcomes. When adjusting for women's characteristics- see Table 21 in S2 Appendix, neonates of primiparous women were significantly more likely to have low birth weight (RR1.85 CI 1.07–3.20), as were neonates of women with high medical risk status as the onset of labour (RR 2.83 CI 1.4305.61). Neonates of women with any social risk factor (RR 2.52 CI 1.02–6.17), and Black Caribbean women (RR11.86 CI 1.23–114.3) were more likely to have a low Apgar score (<8 at 5 minutes), although CI's were wide. Neonatal unit admissions were more likely for Black African women (RR 3.99 CI 1.37–11.64) and those

**Table 7. Use of analgesia in labour and obstetric intervention in relation to the place of antenatal care.**

| Analgesia/ Intervention | Place of antenatal care | Number of women (%) | Unadjusted RR (95% CI) | Model 1 Adjusted RR (95% CI) * | Model 2 Adjusted RR (95% CI) ** | Model 3 Adjusted RR (95% CI) *** |
|---|---|---|---|---|---|---|
| Epidural/CSE/GA | Hospital | 271(57) | 1.33(0.99–1.78) | 1.00(0.71–1.41) | 0.93(0.65–1.33) | 0.90(0.61–1.32) |
| | Community | 203(43) | Ref | Ref | Ref | Ref |
| Opioid analgesia | Hospital | 8(53) | 0.96(0.34–2.68) | 0.93(0.31–2.80) | 1.01(0.29–3.52) | 0.59(0.12–2.93) |
| | Community | 7(47) | Ref | Ref | Ref | Ref |
| No analgesia or Entonox | Hospital | 76(45) | **0.62(0.43–0.87)** | 0.79(0.54–1.16) | 0.87(0.58–1.28) | 1.01(0.66–1.54) |
| | Community | 93(55) | Ref | Ref | Ref | Ref |
| Water in labour | Hospital | 3(23) | **0.24(0.06–0.90)** | 0.28(0.06–1.15) | 0.42(0.09–1.94) | 0.70(0.14–3.52) |
| | Community | 10(77) | Ref | Ref | Ref | Ref |
| CTG in labour | Hospital | 73(26) | **3.88(2.81–5.36)** | **2.83(1.95–4.09)** | **4.18(2.70–6.49)** | 1.08(0.61–1.92) |
| | Community | 210(74) | Ref | Ref | Ref | Ref |
| Induction of labour | Hospital | 171(51) | 0.76(0.57–1.01) | **0.60(0.43–0.84)** | **0.57(0.40–0.80)** | **0.65(0.45–0.95)** |
| | Community | 166(49) | Ref | Ref | Ref | Ref |

* Model 1: Adjustment for demographics ethnicity, age, parity, IMD score, social risk and medical risk factors at booking and onset of labour.

** Model 2: Model 1 + adjustment for model of care.

*** Model 3: Model 2 + adjustment for service provider attended.

with high medical risk status at the onset of labour (RR 4.06 CI 2.10–7.84). Neonatal death and stillbirth was not included in the analysis as numbers in each model of care were too small to detect a relationship (n = 8).

**Analysis 2- place of antenatal care.** Table 11 shows that women receiving antenatal care in the hospital were significantly more likely to have a preterm birth (RR 2.38 CI 1.32–4.27) and neonatal low birth weight (RR 2.31 CI 1.24–4.32) than those receiving antenatal care in

**Table 8. Place of birth in relation to the model of care received.**

| Place of birth | Model of Care | Number of women (%) | Unadjusted RR (95% CI) | Model 1 Adjusted RR (95% CI) * | Model 2 Adjusted RR (95% CI) ** | Model 3 Adjusted RR (95% CI) *** |
|---|---|---|---|---|---|---|
| Birth Centre/midwife led setting | Standard | 119(57) | Ref | Ref | Ref | Ref |
| | Group | 61(29) | Ref | Ref | Ref | Ref |
| | Specialist | 28(14) | Ref | Ref | Ref | Ref |
| Labour ward/obstetric led setting | Standard | 343(62) | **1.64(0.99–2.74)** | 1.41(0.77–2.58) | 1.08(0.58–2.02) | 1.11(0.58–2.12) |
| | Group | 157(29) | 1.47(0.84–2.55) | 1.29(0.67–2.49) | 1.19(0.61–2.31) | 0.76(0.38–1.53) |
| | Specialist | 49(9) | Ref | Ref | Ref | Ref |
| Home | Standard | 2(25) | **0.15(0.02–0.98)** | **0.13(0.01–0.99)** | 0.16(0.01–1.41) | 0.16(0.01–1.45) |
| | Group | 3(38) | 0.45(0.08–2.42) | 0.39(0.05–2.67) | 0.43(0.06–2.99) | 0.34(0.03–2.99) |
| | Specialist | 3(38) | Ref | Ref | Ref | Ref |

* Model 1: Adjusted for demographics ethnicity, age, parity, IMD score, any social and medical risk factors at booking and onset of labour.

** Model 2: Model 1 + Adjustment for place of antenatal care (community or hospital).

*** Model 3: Model 2 + Adjustment for service provider attended (A or B).

**Table 9. Place of birth in relation to place of antenatal care.**

| Place of birth | Place of antenatal care | No of women (%) | Unadjusted RR (95% CI) | Model 1 Adjusted RR (95% CI) * | Model 2 Adjusted RR (95% CI) ** | Model 3 Adjusted RR (95% CI) *** |
|---|---|---|---|---|---|---|
| Birth Centre/midwife led setting | Hospital | 75(36) | Ref | Ref | Ref | Ref |
| | Community | 133(64) | Ref | Ref | Ref | Ref |
| Labour ward/obstetric led setting | Hospital | 340(62) | **2.86(2.05–3.99)** | **2.06(1.40–3.01)** | **2.06(1.38–3.07)** | 1.31(0.85–2.02) |
| | Community | 209(38) | Ref | Ref | Ref | Ref |
| Home | Hospital | 2(25) | 0.59(0.11–3.00) | 0.40(0.06–2.55) | 0.74(0.09–5.60) | 0.58(0.07–4.70) |
| | Community | 6(75) | Ref | Ref | Ref | Ref |

* Model 1: Adjustment for demographics ethnicity, age, parity, IMD score, social risk and medical risk factors at booking and onset of labour.

** Model 2: Model 1 + adjustment for model of care.

*** Model 3: Model 2 + adjustment for service provider attended.

the community setting. These relationships were statistically significant across all models after adjusting for women's characteristics, including their medical risk status, model of care received, and service provider attended. Although no relationship was found between the place of antenatal care and stillbirth or neonatal death, the adjusted analysis presented in Table 22 in S2 Appendix highlight the significance for women with any social risk factor being more likely to have a stillbirth or neonatal death (RR 6.82 CI 1.10–42.15).

## Infant care

**Analysis 1- model of care.** Table 12 shows women were much less likely to have had skin-to-skin contact recorded if they received standard maternity care (RR 0.34 CI 0.14–0.80) and group practice care (RR 0.31 CI 0.13–0.74) compared to those receiving the specialist model. Other women least likely to have had skin-to-skin contact with their infants

**Table 10. Neonatal outcomes in relation to the model of care received.**

| Neonatal outcome | Model of Care | No. of women (%) | Unadjusted RR (95% CI) | Model 1 Adjusted RR (95% CI) * | Model 2 Adjusted RR (95% CI) ** | Model 3 Adjusted RR (95% CI) *** |
|---|---|---|---|---|---|---|
| Gestation <37 weeks at birth | Standard | 52(61) | 1.12(0.51–2.46) | 1.11(0.47–2.62) | 0.81(0.33–1.99) | 0.80(0.33–1.98) |
| | Group | 25(29) | 1.14(0.49–2.66) | 1.08(0.43–2.68) | 0.96(0.38–2.43) | 0.98(0.38–2.50) |
| | Specialist | 8(10) | Ref | Ref | Ref | Ref |
| Birthweight <2500g*** | Standard | 45(63) | 1.59(0.61–4.14) | 1.60(0.57–4.45) | 1.16(0.40–3.36) | 1.16(0.40–3.36) |
| | Group | 21(30) | 1.57(0.57–4.32) | 1.49(0.50–4.36) | 1.24(0.41–3.73) | 1.30(0.43–3.93) |
| | Specialist | 5(7) | Ref | Ref | Ref | Ref |
| Apgar <8 at 5 minutes | Standard | 19(59) | 0.80(0.26–2.42) | 1.44(0.40–5.23) | 1.49(0.40–5.46) | 1.46(0.39–5.37) |
| | Group | 9(28) | 0.80(0.24–2.69) | 1.20(0.30–4.81) | 1.22(0.30–4.84) | 1.42(0.35–5.71) |
| | Specialist | 4(13) | Ref | Ref | Ref | Ref |
| Neonatal unit admission | Standard | 50(61) | 1.78(0.69–4.63) | 1.67(0.60–4.69) | 1.31(0.45–3.80) | 1.31(0.45–3.81) |
| | Group | 27(33) | 2.08(0.77–5.62) | 1.77(0.60–5.22) | 1.58(0.53–4.71) | 1.59(0.53–4.80) |
| | Specialist | 5(6) | Ref | Ref | Ref | Ref |

* Model 1: Adjusted for demographics ethnicity, age, parity, IMD score, any social and medical risk factors at booking and onset of labour.

** Model 2: Model 1 + Adjustment for place of antenatal care (community or hospital).

*** Model 3: Model 2 + Adjustment for service provider attended (A or B).

**Table 11. Neonatal outcomes in relation to the place of antenatal care.**

| Neonatal outcome | Place of antenatal Care | No. of women (%) | Unadjusted RR (95% CI) | Model 1 Adjusted RR (95% CI) * | Model 2 Adjusted RR (95% CI) ** | Model 3 Adjusted RR (95% CI) *** |
|---|---|---|---|---|---|---|
| Gestation <37 at birth | Hospital | 62(73) | **2.45(1.48–4.05)** | **2.18(1.28–3.72)** | **2.26(1.29–3.95)** | **2.38(1.32–4.27)** |
| | Community | 23(27) | Ref | Ref | Ref | Ref |
| Birthweight <2500g* | Hospital | 51(72) | **2.26(1.31–3.87)** | **2.20(1.23–3.92)** | **2.15(1.18–3.92)** | **2.31(1.24–4.32)** |
| | Community | 20(28) | Ref | Ref | Ref | Ref |
| Apgar <8 at 5 minutes | Hospital | 17(53) | 0.89(0.43–1.83) | 0.91(0.42–2.01) | 0.82(0.36–1.85) | 1.25(0.51–3.08) |
| | Community | 15(47) | Ref | Ref | Ref | Ref |
| NNU admission | Hospital | 57(70) | **1.98(1.20–3.26)** | **1.77(1.04–3.02)** | 1.72(0.99–2.99) | 1.74(0.97–3.11) |
| | Community | 25(30) | Ref | Ref | Ref | Ref |
| Neonatal death | Hospital | 4(50) | 0.84(0.20–3.39) | 0.65(0.14–2.90) | 0.60(0.12–2.99) | 0.98(0.20–4.72) |
| | Community | 4(50) | Ref | Ref | Ref | Ref |

* Model 1: Adjusted for demographics: ethnicity, age, parity, IMD score, social risk and medical risk factors at booking and onset of labour.

** Model 2: Model 1 plus adjusted for model of care.

*** Model 3: Model 2 plus Adjusted for service provider attended.

were Black Caribbean women (RR 0.40 CI 0.16–1.00), those with any social risk factor (RR0.59 CI0.38–0.92), women with high medical risk status (RR 0.32 CI 0.21–0.50) and those attending service provider B (RR 0.39 CI 0.22–0.68). No significant relationship was found between model of care and method of infant feeding at discharge from hospital. When the model adjusted for women's characteristics, women with high medical risk status

**Table 12. Feeding method and skin-to-skin in relation to model of care.**

| Feeding method and skin-to-skin | Model of Care | Number of women (%) | Unadjusted RR (95% CI) | Model 1 Adjusted RR (95% CI) * | Model 2 Adjusted RR (95% CI) ** | Model 3 Adjusted RR (95% CI) *** |
|---|---|---|---|---|---|---|
| Breastfeeding at discharge | Standard | 309(61) | Ref | Ref | Ref | Ref |
| | Group | 147(29) | Ref | Ref | Ref | Ref |
| | Specialist | 52(10) | Ref | Ref | Ref | Ref |
| Artificially feeding at discharge | Standard | 32(55) | 1.07(0.40–2.89) | 1.89(0.63–5.63) | 1.69(0.55–5.17) | 1.69(0.55–5.14) |
| | Group | 21(36) | 1.48(0.53–4.14) | 2.10(0.67–6.50) | 2.03(0.65–6.35) | 2.47(0.78–7.78) |
| | Specialist | 5(9) | Ref | Ref | Ref | Ref |
| Mixed Feeding at discharge | Standard | 125(63) | 0.91(0.53–1.55) | 1.10(0.60–1.94) | 1.12(0.61–2.05) | 1.12(0.61–2.04) |
| | Group | 59(25) | 0.75(0.41–1.35) | 0.89(0.47–1.69) | 0.91(0.48–1.72) | 1.16(0.60–2.24) |
| | Specialist | 23(12) | Ref | Ref | Ref | Ref |
| Skin-to-skin | Standard | 348(61) | **0.41(0.20–0.82)** | **0.28(0.12–0.63)** | **0.35(0.15–0.80)** | **0.34(0.14–0.80)** |
| | Group | 156(27) | **0.34(0.16–0.70)** | **0.25(0.10–0.58)** | **0.26(0.11–0.61)** | **0.31(0.13–0.74)** |
| | Specialist | 70(12) | Ref | Ref | Ref | Ref |

* Model 1: Adjusted for demographics ethnicity, age, parity, IMD score, any social and medical risk factors at booking and onset of labour.

** Model 2: Model 1 + Adjustment for place of antenatal care (community or hospital).

*** Model 3: Model 2 + Adjustment for service provider attended (A or B).

**Table 13. Feeding method and skin-to-skin contact in relation to place of antenatal care.**

| Feeding method and skin-to-skin | Place of antenatal care | Number of women (%) | Unadjusted RR (95% CI) | Model 1 Adjusted RR (95% CI) * | Model 2 Adjusted RR (95% CI) ** | Model 3 Adjusted RR (95% CI) *** |
|---|---|---|---|---|---|---|
| Breastfeeding at discharge | Hospital | 276(53) | Ref | Ref | Ref | Ref |
| | Community | 232(46) | Ref | Ref | Ref | Ref |
| Artificially feeding | Hospital | 36(62) | 1.29(0.73–2.27) | 1.45(0.79–2.65) | 1.37(0.73–2.57) | 1.85(0.94–3.65) |
| | Community | 22(38) | Ref | Ref | Ref | Ref |
| Mixed Feeding | Hospital | 104(53) | 0.93(0.67–1.30) | 0.94(0.65–1.35) | 0.90(0.61–1.31) | 1.23(0.82–1.86) |
| | Community | 93(47) | Ref | Ref | Ref | Ref |
| Skin-to-skin | Hospital | 281(49) | **0.42(0.29–0.59)** | **0.52(0.35–0.76)** | **0.53(0.35–0.80)** | 0.69(0.44–1.07) |
| | Community | 293(51) | Ref | Ref | Ref | Ref |

* Model 1: Adjusted for demographics ethnicity, age, parity, IMD score, social risk and medical risk factors.
** Model 2: Model 1 plus adjusted for model of care.
*** Model 3: Model 2 plus Adjusted for service provider attended.

were significantly more likely to be feed their infants artificially (RR2.80 CI 1.33–5.89) or mixed feed (RR 1.78 CI 1.13–2.81). Black Caribbean women were more likely to artificially feed (RR 12.67 CI 1.34–11.8) and those in the Black 'other' ethnic category were more likely to mixed feed (RR 4.26 CI 1.50–12.08).

**Analysis 2- place of antenatal care.**  Table 13 shows that there was no relationship between infant care outcomes and place of antenatal care. For skin-to-skin contact after birth there appeared to be a difference, but when the model adjusted for provider we see that the relationship was driven by women attending service B being less likely to have had skin to skin contact (RR 0.39 CI 0.22–0.68).

## Service use

**Analysis 1- model of care.**  Table 14 shows no significant relationship between the model of care received and the number of antenatal admissions to hospital, or the length of the post-natal stay. However, Black Caribbean (RR 2.86 CI 1.11–7.38) and 'Black other' women (RR 3.59 CI 1.15–11.17) were more likely to have one or more antenatal admissions- see Table 25 in S2 Appendix. Once adjustment was made for provider, women at service B (RR 3.46 CI 1.84–6.50) and those with high medical risk (2.64 CI 1.67–4.18) were more likely to have one or more antenatal admissions, and to stay in hospital after giving birth for 4 or more days (RR 3.91 CI 2.18–7.00).

**Analysis 2- place of antenatal care.**  Table 15 shows no significant relationship between service use outcomes and place of antenatal care.

## Summary of findings

These findings are summarised in Table 16 below, showing the significant findings in relation to either the model of care received, the place of antenatal care, and the service provider. Characteristics of women at disproportionate risk are also presented.

**Table 14. Women's service use in relation to the model of care received.**

| Service use | Model of care | Number of women (%) | Unadjusted RR (95% CI) | Model 1 Adjusted RR (95% CI) * | Model 2 Adjusted RR (95% CI) ** | Model 3 Adjusted RR (95% CI) *** |
|---|---|---|---|---|---|---|
| 1 or more antenatal admissions | Standard | 90(60) | 1.02(0.56–1.88) | 1.04(0.52–2.07) | 0.90(0.44–1.84) | 0.89(0.43–1.86) |
| | Group | 46(30) | 1.13(0.59–2.17) | 1.10(0.53–2.29) | 1.07(0.51–2.25) | 0.81(0.37–1.76) |
| | Specialist | 15(10) | Ref | Ref | Ref | Ref |
| Length of postnatal stay: | | | | | | |
| 0–1 day | Standard | 227(60) | Ref | Ref | Ref | Ref |
| | Group | 116(30) | Ref | Ref | Ref | Ref |
| | Specialist | 36(10) | Ref | Ref | Ref | Ref |
| 2 days | Standard | 118(66) | 0.93(0.51–1.68) | 0.88(0.45–1.70) | 0.85(0.43–1.67) | 0.85(0.43–1.68) |
| | Group | 42(23) | 0.65(0.34–1.24) | 0.60(0.29–1.24) | 0.60(0.29–1.23) | 0.52(0.25–1.11) |
| | Specialist | 20(11) | Ref | Ref | Ref | Ref |
| 3 days | Standard | 54(57) | 0.85(0.40–1.83) | 0.84(0.36–1.93) | 0.90(0.38–2.13) | 0.91(0.38–2.14) |
| | Group | 30(32) | 0.93(0.41–2.08) | 0.83(0.34–2.00) | 0.85(0.35–2.06) | 0.86(0.35–2.14) |
| | Specialist | 10(11) | Ref | Ref | Ref | Ref |
| 4 or more days | Standard | 70(60) | 0.79(0.40–1.55) | 0.61(0.29–1.31) | 0.61(0.27–1.34) | 0.61(0.28–1.35) |
| | Group | 33(28) | 0.73(0.35–1.51) | 0.63(0.27–1.42) | 0.63(0.27–1.43) | 0.72(0.31–1.65) |
| | Specialist | 14(12) | Ref | Ref | Ref | Ref |

* Model 1: Adjusted for demographics ethnicity, age, parity, IMD score, any social and medical risk factors at booking and onset of labour.
** Model 2: Model 1 + Adjustment for place of antenatal care (community or hospital).
*** Model 3: Model 2 + Adjustment for service provider attended (A or B).

## Subgroup analysis

Outcomes that were associated with a significant relationship to either the model of care received, or the place of antenatal care attended were analysed for the 'most at risk' women only. This subgroup included:

**Table 15. Women's service use in relation to place of antenatal care.**

| Service use | Place of antenatal care | Number of women (%) | Unadjusted RR (95% CI) | Model 1 Adjusted RR (95% CI) * | Model 2 Adjusted RR (95% CI) ** | Model 3 Adjusted RR (95% CI) *** |
|---|---|---|---|---|---|---|
| 1 or more antenatal admissions | Hospital | 102(68) | **2.00(1.37–2.91)** | 1.38(0.91–2.09) | 1.46(0.94–2.26) | 1.00(0.61–1.64) |
| | Community | 49(32) | Ref | Ref | Ref | Ref |
| Length of postnatal stay: | | | | | | |
| 0–1 day | Hospital | 199(52) | Ref | Ref | Ref | Ref |
| | Community | 180(48) | Ref | Ref | Ref | Ref |
| 2 days | Hospital | 105(58) | 1.25(0.87–1.80) | 1.15(0.77–1.71) | 1.09(0.72–1.67) | 0.93(0.58–1.48) |
| | Community | 75(42) | Ref | Ref | Ref | Ref |
| 3 days | Hospital | 49(52) | 0.97(0.62–1.53) | 0.83(0.50–1.37) | 0.83(0.49–1.40) | 0.83(0.47–1.46) |
| | Community | 45(48) | Ref | Ref | Ref | Ref |
| 4 or more days | Hospital | 65(56) | 1.08(0.71–1.65) | 0.85(0.60–1.52) | 0.98(0.60–1.61) | 1.14(0.68–1.93) |
| | Community | 52(44) | Ref | Ref | Ref | Ref |

* Model 1: Adjusted for demographics ethnicity, age, parity, IMD score, social risk and medical risk factors.
** Model 2: Model 1 plus adjusted for model of care.
*** Model 3: Model 2 plus Adjusted for service provider attended.

**Table 16. Overview of outcomes.**

| Outcome variable | Characteristics of women at disproportionate risk when adjusting (S2 Appendix) | Significant effect of specialist model of care | Significant effect of hospital based antenatal care | Significant effect of service provider |
|---|---|---|---|---|
| **Maternal birth outcomes** | | | | |
| Elective caesarean section | High medical risk | = | = | B ↑ |
| Emergency caesarean section | Primiparous | = | = | = |
| | High medical risk | | | |
| Instrumental delivery | Primiparous | = | = | = |
| | High medical risk | | | |
| Postpartum haemorrhage | Primiparous | = | = | B ↑ |
| | High medical risk | | | |
| Massive obstetric haemorrhage | High medical risk | = | = | = |
| | Social risk factor(s) | | | |
| Perineal trauma | Primiparous | = | = | = |
| Obstetric emergency | Primiparous | = | = | = |
| | High medical risk | | | |
| Epidural/CSE/GA in labour | Primiparous | = | = | B ↑ |
| | High medical risk | | | |
| | Over 34 years old | | | |
| Opioid in labour | Primiparous | = | = | = |
| No analgesia or Entonox only in labour | Multiparous | = | = | = |
| Water for pain relief in labour | High medical risk | ↑ | = | = |
| | Increased age | | | |
| Monitoring (CTG in labour) | Primiparous | = | = | B ↑ |
| | High medical risk | | | |
| Induction of labour | Primiparous | = | ↓ | = |
| | High medical risk | | | |
| Place of birth- obstetric led | High medical risk | = | = | B ↑ |
| **Neonatal Outcomes** | **Characteristics of women at disproportionate risk when adjusting (S2 Appendix)** | **Significant effect of specialist model of care** | **Significant effect of hospital based antenatal care** | **Significant effect of service** |
| Premature birth (<37/40weeks) | Primiparous | = | ↑ | = |
| Low birthweight (<2500g) | Primiparous | = | ↑ | = |
| | High medical risk | | | |
| Apgar scores | Social risk factor(s) | = | = | B↓ |
| | Black Caribbean | | | |
| Neonatal unit admission | Black African | = | = | = |
| Stillbirth/neonatal death | Social risk factor(s) | N/A | = | = |
| Artificially fed infant at discharge | High medical risk | = | = | B ↑ |
| | Black 'other' ethnicity | | | |
| Skin-to-skin contact | Black Caribbean | ↑ | = | B ↓ |
| | Social risk factor(s) | | | |
| | High medical risk | | | |
| **Hospital stay** | | | | |
| Antenatal admissions | Black Caribbean | = | = | B ↑ |
| | Black 'other' | | | |
| | High medical risk | | | |

*(Continued)*

**Table 16.** (Continued)

| Length of postnatal stay | Primiparous | = | = | = |
|---|---|---|---|---|
| | High medical risk | | | |

↑ = Statistically significant increase (Pr < 0.05).

↓ = Statistically significant decrease (Pr < 0.05).

= No significant relationship detected.

'A' and 'B' refer to services.

- Women with IMD scores within the most deprived 3 deciles and/or

- Not white ethnicity and/or

- Any social risk factor

This subgroup accounted for 593 women, 59.30% of the sample.

**Analysis 1- model of care.** Table 17 shows that of the 593 women with increased social risk, only 7 used water for pain relief in labour. The relationship to model of care was not seen for women with increased social risk. However, for skin-to-skin contact there remained a significant relationship, with women at increased risk who received the specialist model of care being more likely to experience this important bonding practice.

**Analysis 2- place of antenatal care.** The significant outcomes associated with place of antenatal care were also analysed for the 'most at risk' subgroup, see Table 18. When the rate of induction of labour was analysed for the subgroup the relationship between hospital and increased induction was no longer significant suggesting that the women with less social risk attending hospital antenatal care are more likely to experience induction of labour. For the whole sample, women attending the hospital for their antenatal care are more likely to experience preterm birth compared to those attending community based antenatal care (RR2.38 CI 1.32–4.27), but the risk increases for the 'most at risk' subgroup (RR 3.11 CI1.49–6.50). The relationship between hospital-based antenatal care and low birthweight remained significant but did not increase for the subgroup.

Further findings of the wider evaluation can be found at https://www.project20.uk.

**Table 17. Subgroup analysis by model of care received.**

| Outcome | Model of Care | Number of women (%) | Unadjusted RR (95% CI) | Model 1 Adjusted RR (95% CI) * | Model 2 Adjusted RR (95% CI) ** | Model 3 Adjusted RR (95% CI) *** |
|---|---|---|---|---|---|---|
| Water for pain relief in labour | Standard | 3(43) | 0.61(0.63–6.05) | 0.55(0.41–7.50) | 0.64(0.04–8.85) | 0.45(0.02–6.98) |
| | Group | 3(43) | 1.35(0.12–13.3) | 1.61(0.12–20.4) | 1.67(0.13–21.3) | 1.90(0.14–25.3) |
| | Specialist | 1(14) | Ref | Ref | Ref | Ref |
| Skin-to-skin contact | Standard | 216(59) | **0.43(0.21–0.89)** | **0.28(0.13–0.64)** | **0.37(0.16–0.85)** | **0.32(0.13–0.77)** |
| | Group | 97(26) | **0.39(0.18–0.85)** | **0.28(0.12–0.65)** | **0.29(0.12–0.69)** | **0.36(0.14–0.89)** |
| | Specialist | 54(15) | Ref | Ref | Ref | Ref |

* Model 1: Adjusted for demographics ethnicity, age, parity, IMD score, any social and medical risk factors at booking and onset of labour.

** Model 2: Model 1 + Adjustment for place of antenatal care (community or hospital).

*** Model 3: Model 2 + Adjustment for service provider attended (A or B).

**Table 18. Subgroup analysis by place of antenatal care attended.**

| Outcome | Place of antenatal care | Number of women (%) | Unadjusted RR (95% CI) | Model 1 Adjusted RR (95% CI) * | Model 2 Adjusted RR (95% CI) ** | Model 3 Adjusted RR (95% CI) *** |
|---|---|---|---|---|---|---|
| Induction of labour | Hospital | 118(52) | 0.86(0.60–1.23) | 0.92(0.63–1.34) | 0.88(0.60–1.31) | 0.87(0.55–1.37) |
| | Community | 104(46) | Ref | Ref | Ref | Ref |
| Preterm Birth | Hospital | 45(76) | **2.85(1.52–5.35)** | **3.15(1.62–6.15)** | **3.24(1.63–6.42)** | **3.11(1.49–6.50)** |
| | Community | 14(24) | Ref | Ref | Ref | Ref |
| Low birthweight | Hospital | 35(67) | 1.76(0.95–3.23) | **2.21(1.14–4.30)** | **2.10(1.06–4.15)** | **2.09(1.00–4.34)** |
| | Community | 17(33) | Ref | Ref | Ref | Ref |

* Model 1: Adjusted for demographics ethnicity, age, parity, IMD score, social risk and medical risk factors.

** Model 2: Model 1 plus adjusted for model of care.

*** Model 3: Model 2 plus Adjusted for service provider attended.

## Discussion

Although there is encouraging evidence that continuity of midwifery care improves birth outcomes [40], the mechanisms underlying these improvements, and the impact on clinical outcomes for women with social risk factors is largely absent in the literature, prompting a need for this research. The two specialist models of care evaluated in this study were similar in that they both provided continuity of midwifery care to women with low socioeconomic status and social risk factors. The main differences between the models is that one was based within a local community health service 'hub' and the other within a large teaching hospital. These differences allowed for the exploration of mechanisms based not only on continuity of care but also the impact of place-based care.

To summarise, no significant differences were found between the model of care and majority of the maternal outcomes. This is important considering women in the specialist model were more likely to have high deprivation scores and more social risk factors, and therefore more likely to experience poor maternal birth outcomes such as caesarean section and obstetric emergencies [4, 27, 68, 69]. Therefore the specialist model of care, and in some cases the group practice model, appear to offer protection against the poorer outcomes that might be expected for these women under standard care'. The specialist model of care was associated with improved birth outcomes such as skin to skin contact after birth and the use of water as pain relief in labour. Interestingly, different relationships were found between the place of antenatal care and neonatal outcomes such as premature birth and low birth weight. Table 16 presents specific demographics that put women at disproportionate risk of each outcome. These were often related to race, age, parity, deprivation score, medical risk status, and social risk factors, and were used to develop a subgroup to further analyse the significant findings. Social risk factors were associated with an increase in stillbirth and neonatal death in the adjusted models. Given the small numbers and findings for preterm births and neonatal unit admissions this warrants further investigation of the relationship between place of antenatal care and stillbirth or neonatal death in future research. The subgroup analysis found that for most outcomes there was little difference in effect compared to the whole cohort, but when preterm birth was analysed for the sub-group, women attending the hospital-based model who were at increased social risk were even more likely to have premature birth than those in the full analysis.

The variation seen between different aspects of maternity care, women's demographics and their outcomes highlights the individual nature of pregnancy and birth. There is no 'one size fits all' approach to improving all outcomes for all women; care must be tailored to meet these individual needs and a starting point for this is continuity of care through which women's needs can be realised. The differences seen between the findings of this study and the Cochrane review of midwife led models of care [40] can be largely explained by the population being analysed and the place of care. Where some of the trials included in the review excluded women with medical risk factors and substance abuse, others were based in the hospital setting. A subgroup analysis of women with social risk factors and place of antenatal care would be a useful contribution to the review.

As established in the introduction, women with low socioeconomic status and social risk factors are more susceptible to poor infant birth outcomes, including preterm birth (birth before 37 weeks' gestation). Despite efforts to decrease its prevalence, improve clinical management and reduce infant morbidity and mortality, preterm birth rates continue to rise in most countries [70]. This is an important outcome and indicator for health over the life course with many preterm neonates going on to have significant developmental delay, learning disabilities, visual and hearing problems, chronic lung disease as well as other health implications [71, 72]. These factors lead to increased costs to health services, the economy and the broader society [70]. There are many predisposing, and often intersecting, factors associated with preterm birth that are important to bear in mind as we attempt to propose specific mechanisms that reduce preterm birth rates for women who are accessing care in the community setting. These factors include; infection, social stress, intimate partner violence, non-Caucasian ethnicity, young or advanced age, previous preterm birth, short inter-pregnancy intervals, nutritional deficiencies, cervical procedures, underlying medical conditions, smoking and alcohol consumption, and pollution exposure [31, 73–75]. As discussed in the introduction, women from Black and minority ethnic groups and those with social risk factors are more likely to be living in poverty, experiencing multiple health issues and have poorer experiences of healthcare services driven by discrimination than their white counterparts [76, 77]. Women from Black and minority ethnic groups, who are at increased risk of preterm birth, have described the effect of 'weathering' in relation to accessing medical care. First coined by Geronimus [78], 'weathering' posits that Black women's health deteriorates in early adulthood as a result of the cumulative effects of socioeconomic disadvantage. The theory has been widely tested and supported through analysis of health inequalities seen in pregnancy outcomes, excess mortality, disability and mental health [79–83]. Geronimus' theory led the way to phenomena such as the allostatic load [84], epigenetics [85], and telomere shortening (a marker of cellular aging), all of which have been associated with preterm birth and the cumulative effect of stress and exposure to discrimination on the body [86–88]. The question is then, how can maternity care acknowledge and aim to reduce the effect of these stressors, not only to improve pregnancy outcomes but also to break the cycle of socioeconomic disadvantage and its associated health inequalities?

In addition to the Cochrane reviews of models of care and interventions to reduce preterm birth [40, 89], a systematic review and meta-analysis of models of antenatal care designed to reduce preterm birth [61] found that women randomised to midwife-led continuity models of antenatal care were less likely to experience preterm birth regardless of their medical risk factors. The evidence base concludes that although alternative models of antenatal care can be effective in reducing preterm birth, the mechanisms, including the effect of place of antenatal care remain unknown.

The hospital environment has long been associated with increased stress, waiting times, unfamiliarity, fragmentation and impersonal care [30, 90–92]. When the stressful effects of the

hospital environment are compounded by paternalistic care, a lack of choice and perceived stigma and discrimination often described by Black and minority ethnic women and those with social risk factors [28, 30, 93], poor outcomes and experiences can be exacerbated. Acknowledgement of the effect of environment is seen in recent policy with the NHS long term plan [94] and five year forward view [95] emphasising the value of expanding community-based health services on people's health, help-seeking behaviours and pressures on the wider service. The concept of women handing over control and choice to the healthcare professional was also described in Ebert et al.'s [96] qualitative work with socially disadvantaged women in Australia. The study concluded with the recommendation to step away from medically focused maternity care environments in order to create 'safe spaces' for women. Although the place of care is not discussed in Ebert et al.'s study, care set within women's local communities could be a solution to protecting women from the medicalised hospital environment where they feel disempowered and silenced. Supporting this theory, focus group's with midwives providing the specialist models of care evaluated in this study described how midwives working in the community setting were more sensitive to women's wider needs, able to act quickly on abnormal findings or concerns and had increased knowledge of local support available. Another protective factor of community-based care to consider is that of 'ethnic maintenance', describing the social connections and cultural norms that multi-cultural communities provide [97]. These combined insights contribute to the theory that care based within the community setting may be perceived by women to align more closely with their needs as the service has 'come to them'. These insights are presented in 'context + mechanism = outcome' format in Fig 2:

This study also found significant relationships between the specialist model of care and the increased number of women using water for pain relief and practising skin-to-skin contact with their baby shortly after birth. The use of water for non-pharmacological pain relief in labour is associated with a reduction in the duration of labour and use of epidural anaesthesia, fewer interventions and transfer to obstetric units in labour, and no adverse outcomes [98, 99]. Skin-to-skin contact, sometimes referred to as 'kangaroo care', can be defined as 'placing a naked infant onto the bare chest of the mother' [100] the benefits of which include improved adaptation to extrauterine life, stimulation of the digestive system and hormone release leading to improved feeding, protection against infection, reduced cortisol levels, and parent-infant bonding [100–103]. A recently published trial of a specialist continuity model of care for women at risk of preterm birth found those women randomised to the intervention were significantly more likely to have skin-to-skin contact after birth and to have it for a longer time [104]. Although the underlying mechanisms for these outcomes remain unclear and warrant further research, the phenomenon could be explained using Lipsky's [105] street level bureaucracy theory in that when midwives know women and are invested in them and their outcomes, they are more likely to provide gold standard practice. This was referred to in the focus groups with the midwives providing the specialist models of care [106] and is summed up well in the following quote: *"I think we also have that like emotional insight as well. . . I feel like we, as a team, we are quite invested in our women, and we do do a lot for them and I think, when you have that investment in someone that you want to push for them, and you want their outcome to be good'.*

## Strenghts and limitations

As with all cohort studies, particularly those using records that were not designed for the purpose of the study, there are limitations such as potential poor data quality and differential loss

**Context**

Community based antenatal care in area of social deprivation providing care for women with low socioeconomic status and social risk factors

These women are more likely to experience discrimination, paternalistic and depersonalised care and are at a higher risk of poor birth outcomes such as preterm birth

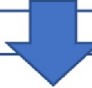

**Resource mechanisms**

Continuity of care (more appointments with a known healthcare professional and more likely to be looked after in labour by a known HCP)

Reduced travel costs and inconvenience

Family friendly environment where women are able to bring children without fear of reproach

Familiar environment, processes (for booking appointments, following up test results etc) and faces (reception staff, healthcare professionals)

Community based antenatal education that includes information relevant to local populations

Referring, escalating concerns, and handing over information to relevant professionals in the same local setting to ensure women do not fall through the gaps

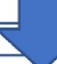

**Response mechanisms**

Trust in healthcare professional and perceived 'safe space'

Help-seeking behaviour and disclosure of social risk factors

Engagement with services as an active, well informed participant

Referral to specialist services to meet women's physical, social and emotional needs

Social capital and opportunity to meet local women and form support networks

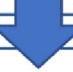

**Outcome**

Appropriate, needs led care where concerns are raised in a timely manner and healthcare professionals respond with empathy and evidence-based practice.

Reduced stress, anxiety, experiences of paternalistic care and discrimination

Established support groups and professional networks for the early years

Early detection of abnormalities and appropriate care plans put in place

Improved outcomes (including preterm birth) and experiences of care

**Fig 2. CMO configuration to reduce inequalities seen in preterm birth rates.**

to follow up [107]. In an attempt to minimise these limitations we took a prospective approach to data collection and analysed the demographic data of women who did not go on to give birth at the study sites. The analysis of maternal and neonatal birth outcomes took a multi-nominal model approach to separately analyse the effect of numerous potentially confounding

factors such as demographics, the place of care, model of care received, and service attended. Limited by the use of the IMD score as the only measure of deprivation available to the researchers, future research would be strengthened by analysing other potentially confounding factors and measures of deprivation and social exclusion such as income, occupation, and women's support networks, as well as measures of perceived discrimination that has been linked to poorer maternal outcomes in the US [108]. An analysis of maternal deaths in the UK [16] found medical comorbidities to be a main driver of maternal death. The analysis did not include mental health as a comorbidity but more recent research [109] highlighted it's significant impact on maternal morbidity. Although this study adjusted for high medical risk factors, data on the number and nature of medical conditions was not collected and should be addressed in future research to identify the underlying mechanisms and how health services can better meets the needs of those with physical and mental health comorbidities. Differences in the models of care and place of antenatal care such as working practises, environment and midwives characteristics could also refine our understanding of the causal mechanisms leading to improved outcomes. The relatively small numbers in each quantitative data group should be taken into consideration due to the significant amount of multiple testing required to establish the separate effects of the potentially confounding factors. This presents a potential limitation as the use of multiple testing can result in a substantial change statistical power, reducing the probability of detecting effects when they do exist and increasing the probability of finding significant differences by chance [110]. This could be overcome in future research using larger sample sizes to test the apparent mitigating effects of the specialist models and community antenatal care on health inequalities.

The generalisability of the findings is limited by the urban location of both specialist models of care evaluated. This is particularly significant when reflecting on the outcomes relating to place-based care- what may have significant outcomes in a densely populated, inner-city, multicultural community, may yield very different results elsewhere. Research is needed to test the generalisability of the findings to rural and other community settings. The wider evaluation of the models of care described in this study will give insight into the underlying mechanisms for the outcomes reported, as well as further detail into women's access and engagement and the quality of the relational continuity they experienced.

## Conclusion

The findings presented in this study highlight how different aspects of maternity care can lead to different outcomes dependant on women's specific demographics and circumstances. It reveals insight into the complexity of the mechanisms underpinning specialist models of care and how they could lead to a narrowing of inequalities in pregnancy related outcomes. Unpicking these mechanisms allows the formation of new hypotheses to test around place-based care, its impact on neonatal outcomes, and the development of maternity services that aim to reduce health inequalities for local populations. The mechanisms listed in the CMO configuration could be used as a framework for those developing maternity services, with the recommendation to audit outcomes to enable a greater understanding of how they might work in different contexts. Although the findings support the policy drive to scale up models of maternity care that offer continuity to those at increased risk, this study reveals that continuity alone is not a panacea for all pregnancy and birth inequalities. Other aspects of maternity care such as where it is placed, the level of choice and control it provides women with, and how autonomously midwives can practice need to be carefully considered by those implementing services.

## Supporting information

**S1 Appendix. Definitions.**
(DOCX)

**S2 Appendix. Data analysis.**
(DOCX)

## Acknowledgments

The authors would like to thank student midwives Laura Peazold, Mary Newman, Natalie Goodyear and Micaela Anthony for help with data collection and anonymisation.

## Author Contributions

**Conceptualization:** Hannah Rayment-Jones, James Harris, Jane Sandall.

**Data curation:** Hannah Rayment-Jones, Elidh Parslow, Thomas Georgi.

**Formal analysis:** Hannah Rayment-Jones, Kathryn Dalrymple, Angela Harden, Jane Sandall.

**Funding acquisition:** Hannah Rayment-Jones.

**Investigation:** Hannah Rayment-Jones, Elidh Parslow, Thomas Georgi, Jane Sandall.

**Methodology:** Hannah Rayment-Jones, Kathryn Dalrymple, Angela Harden, Jane Sandall.

**Project administration:** Hannah Rayment-Jones.

**Resources:** Hannah Rayment-Jones.

**Supervision:** James Harris, Angela Harden, Jane Sandall.

**Validation:** Kathryn Dalrymple, Jane Sandall.

**Visualization:** Hannah Rayment-Jones.

**Writing – original draft:** Hannah Rayment-Jones.

**Writing – review & editing:** Hannah Rayment-Jones, Kathryn Dalrymple, James Harris, Angela Harden, Elidh Parslow, Thomas Georgi, Jane Sandall.

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
