## [Decision Letter · Decision Letter 0]

1 Mar 2021

PONE-D-20-40920

Project20: What aspects of maternity care improve maternal and neonatal birth outcomes for women with social risk factors? A prospective, observational study

PLOS ONE

Dear Dr. Rayment-Jones,

Thank you for submitting your manuscript to PLOS ONE. After careful consideration, we feel that it has merit but does not fully meet PLOS ONE’s publication criteria as it currently stands. Therefore, we invite you to submit a revised version of the manuscript that addresses the points raised during the review process.

We look forward to receiving your revised manuscript.

Kind regards,

Hannah Dahlen, RN, RM, BN (Hons), MCommN, PhD FACM

Academic Editor

PLOS ONE

Journal Requirements:

3.We note that the grant information you provided in the ‘Funding Information’ and ‘Financial Disclosure’ sections do not match.

4.Thank you for stating the following in your Competing Interests section: 

"No "

6. Please include a caption for figure 2.

Additional Editor Comments:

Thank you for this interesting paper. It is clear the reviewers think it a very important and timely contribution to knowledge. Please address the concerns Reviewer One has about the design and analysis and provide more information.

Reviewers' comments:

Reviewer's Responses to Questions

**Comments to the Author**

1. Is the manuscript technically sound, and do the data support the conclusions?

Reviewer #1: Partly

Reviewer #2: Yes

2. Has the statistical analysis been performed appropriately and rigorously? 

Reviewer #1: I Don't Know

Reviewer #2: I Don't Know

3. Have the authors made all data underlying the findings in their manuscript fully available?

Reviewer #1: Yes

Reviewer #2: Yes

4. Is the manuscript presented in an intelligible fashion and written in standard English?

Reviewer #1: No

Reviewer #2: Yes

5. Review Comments to the Author

Reviewer #1: This is an important paper in an area that will be of great interest to the readership of the journal. However the study design is problematic (as with all observational cohort studies) and insufficient detail has been provided of the limitations of the design- the description of the analysis is obscure to say the least and it would be great to see a plain English description that aims to make it very transparent to the readers what exactly was done and why they should trust this was done appropriately. Could comment also be made about how to position these findings and what the implications are for future practice with this cohort of women. What was not assessed in this cohort that might have made a difference?

Reviewer #2: This is an excellent paper, very timely and addressing an area much in need of research. The introduction is appropriate and addresses key issues related to social factors impacting on the lives and outcomes of pregnant women and their children.

The methods are clearly described and embedded in the larger study

Results are clearly laid out and explained based on the models tested.

The discussion addresses the key findings.

There is one sentence in the discussion that is not complete "In addition to the Cochrane review of midwifery led models of care 40, a systematic review and metaanalysis of models of antenatal care designed to prevent and reduce preterm birth 62.... does not lead to a conclusion in this sentence.

Thank you for the opportunity to review this excellent paper.

6. PLOS authors have the option to publish the peer review history of their article (what does this mean?). If published, this will include your full peer review and any attached files.

Reviewer #1: **Yes: **Maralyn Foureur

Reviewer #2: No

---

## [Author Response · Author response to Decision Letter 0]

24 Mar 2021

Reviewer #1: This is an important paper in an area that will be of great interest to the readership of the journal. 

Response: Thank you for your kind and encouraging comments. Please see the revisions made in response to your suggestions below: 

Reviewer #1:The study design is problematic (as with all observational cohort studies) and insufficient detail has been provided of the limitations of the design.

Response: Added to ‘Strengths and limitations’ section of discussion: ‘As with all cohort studies, particularly those using records that were not designed for the purpose of the study, there are limitations such as potential poor data quality and differential loss to follow up 108. In an attempt to minimise these limitations we took a prospective approach to data collection and analysed the demographic data of women who did not go on to give birth at the study sites. The analysis of maternal and neonatal birth outcomes took a multinominal model approach to separately analyse the effect of numerous potentially confounding factors such as demographics, the place of care, model of care received, and service attended. 

Reviewer #1: The description of the analysis is obscure to say the least and it would be great to see a plain English description that aims to make it very transparent to the readers what exactly was done and why they should trust this was done appropriately. 

Response: The title has been changed to clarify what was tested in this study from the offset. The title is now: Project20: Does continuity of care and community-based antenatal care improve maternal and neonatal birth outcomes for women with social risk factors? A prospective, observational study

Added to author summary: ‘In order to understand the extent to which women’s characteristics, the model of care they accessed, where they received their antenatal care and what service they used had an effect on their birth outcomes, we followed a commonly used statistical technique to analyse each and report the effect size.’ 

Added to analysis section: ‘Three regression models were developed to identify the differences in the effect size for each: Model 1 adjusted for ethnicity, age, parity, deprivation score, social risk factors and medical risk status, Model 2 included model 1, plus adjustment for the service provider that women attended to consider differences in organisation guidelines, processes and culture and Model 3 included model 2, however, the place of antenatal care (hospital versus community-based antenatal care) was treated as the independent variable.’

Reviewer #1:Could comment also be made about how to position these findings and what the implications are for future practice with this cohort of women. 

Response: Added to conclusion in abstract: . ‘The findings support the current policy drive to increase continuity of midwife-led care, whilst adding that community-based care may further improve outcomes for women at increased risk of health inequalities’

Added to conclusion: ‘The mechanisms listed in the CMO configuration could be used as a framework for those developing maternity services, with the recommendation to audit outcomes to enable a greater understanding of how they might work in different contexts.’ 

Reviewer #1:What was not assessed in this cohort that might have made a difference?

Response: Added to ‘Strengths and limitations’ section of discussion: ‘Limited by the use of the IMD score as the only measure of deprivation available to the researchers, future research would be strengthened by analysing other potentially confounding factors and measures of deprivation and social exclusion such as income, occupation, and women’s support networks, as well as measures of perceived discrimination that has been linked to poorer maternal outcomes in the US 109. An analysis of maternal deaths in the UK16 found medical comorbidities to be a main driver of maternal death. The analysis did not include mental health as a comorbidity but more recent research 110 highlighted it’s significant impact on maternal morbidity. Although this study adjusted for high medical risk factors, data on the number and nature of medical conditions was not collected and should be addressed in future research to identify the underlying mechanisms and how health services can better meets the needs of those with physical and mental health comorbidities. Differences in the models of care and place of antenatal care such as working practises, environment and midwives characteristics could also refine our understanding of the causal mechanisms leading to improved outcomes.’

Reviewer #2: This is an excellent paper, very timely and addressing an area much in need of research. The introduction is appropriate and addresses key issues related to social factors impacting on the lives and outcomes of pregnant women and their children.

The methods are clearly described and embedded in the larger study

Results are clearly laid out and explained based on the models tested.

The discussion addresses the key findings.

There is one sentence in the discussion that is not complete "In addition to the Cochrane review of midwifery led models of care 40, a systematic review and metaanalysis of models of antenatal care designed to prevent and reduce preterm birth 62.... does not lead to a conclusion in this sentence.

Thank you for the opportunity to review this excellent paper.

Response: Thank you for your time reviewing this paper and the thoughtful comments. 

Revised as: In addition to the Cochrane reviews of models of care and interventions to reduce preterm birth 40,91, a systematic review and meta-analysis of models of antenatal care designed to reduce preterm birth 62 found that women randomised to midwife-led continuity models of antenatal care were less likely to experience preterm birth regardless of their medical risk factors.

---

## [Decision Letter · Decision Letter 1]

19 Apr 2021

Project20: Does continuity of care and community-based antenatal care improve maternal and neonatal birth outcomes for women with social risk factors? A prospective, observational study

PONE-D-20-40920R1

Dear Hannah

We’re pleased to inform you that your manuscript has been judged scientifically suitable for publication and will be formally accepted for publication once it meets all outstanding technical requirements.

Kind regards,

Hannah Dahlen, RN, RM, BN (Hons), MCommN, PhD FACM

Academic Editor

PLOS ONE

Additional Editor Comments (optional):

Thanks for addressing all the reviewers comments

Reviewers' comments:

Reviewer's Responses to Questions

**Comments to the Author**

1. If the authors have adequately addressed your comments raised in a previous round of review and you feel that this manuscript is now acceptable for publication, you may indicate that here to bypass the “Comments to the Author” section, enter your conflict of interest statement in the “Confidential to Editor” section, and submit your "Accept" recommendation.

Reviewer #1: All comments have been addressed

Reviewer #2: All comments have been addressed

2. Is the manuscript technically sound, and do the data support the conclusions?

Reviewer #1: Yes

Reviewer #2: Yes

3. Has the statistical analysis been performed appropriately and rigorously? 

Reviewer #1: Yes

Reviewer #2: I Don't Know

4. Have the authors made all data underlying the findings in their manuscript fully available?

Reviewer #1: Yes

Reviewer #2: Yes

5. Is the manuscript presented in an intelligible fashion and written in standard English?

Reviewer #1: Yes

Reviewer #2: Yes

6. Review Comments to the Author

Reviewer #1: Thank you for addressing the previous comments carefully- I hope the readers enjoy the paper as much as I do now!

Reviewer #2: Thank you for the chance to look at the revisions to the paper. Considerable effort has been pput into addressing the concerns of the reviewers.

I have no further comments

7. PLOS authors have the option to publish the peer review history of their article (what does this mean?). If published, this will include your full peer review and any attached files.

Reviewer #1: **Yes: **Professor Maralyn Foureur

Reviewer #2: No

---

## [Editor Report · Acceptance letter]

22 Apr 2021

PONE-D-20-40920R1 

Project20: Does continuity of care and community-based antenatal care improve maternal and neonatal birth outcomes for women with social risk factors? A prospective, observational study 

Dear Dr. Rayment-Jones:

I'm pleased to inform you that your manuscript has been deemed suitable for publication in PLOS ONE. Congratulations! Your manuscript is now with our production department. 

Kind regards, 

on behalf of

Dr. Hannah Dahlen 

Academic Editor

PLOS ONE